# Specific Patterns of Blood ILCs in Metastatic Melanoma Patients and Their Modulations in Response to Immunotherapy

**DOI:** 10.3390/cancers13061446

**Published:** 2021-03-22

**Authors:** Louise Rethacker, Marie Roelens, Claudia Bejar, Eve Maubec, Hélène Moins-Teisserenc, Anne Caignard

**Affiliations:** 1INSERM UMRS1160, Institut de Recherche Saint Louis, Hôpital Saint-Louis, 1 Avenue Claude Vellefaux, 75010 Paris, France; louise.rethacker@inserm.fr (L.R.); marie.roelens@aphp.fr (M.R.); 2Dermatology department, AP-HP Hôpital Avicenne and University Paris 13, 93008 Bobigny, France; claudia.bejar@aphp.fr (C.B.); eve.maubec@aphp.fr (E.M.); 3Institut de Recherche Saint-Louis, AP-HP hopital Saint-Louis, Université de Paris, INSERM UMRS-1160, 75010 Paris, France; helene.moins@inserm.fr

**Keywords:** melanoma, innate lymphoid cells, ipilimumab, natural killer cells

## Abstract

**Simple Summary:**

Anti-CTLA-4 and anti-PD-1 immune checkpoints inhibitors (ICI) have revolutionized the treatment of metastatic melanoma patients, leading to durable responses. However, some patients still not respond to this clinically used immunotherapies and there is a lack of biomarkers leading to the choice of first-line therapies. Innate lymphoid cells (ILC) express immune checkpoint receptors and are involved in anti-melanoma immune response. The aim of this article is to study ILCs from peripheral blood of melanoma patients receiving Ipilimumab, an anti-CTLA-4 treatment, and their association with clinical responses to this therapy. Our results show an impact of Ipilimumab on ILCs proportions and phenotype in blood. Moreover, the presence of anergic CD56^dim^CD16^−^DNAM-1^−^ NK cells were associated with progression of the disease. These findings demonstrate the important role of ILC in the response to ICI.

**Abstract:**

Immunotherapy targeting immune checkpoint receptors brought a breakthrough in the treatment of metastatic melanoma patients. However, a number of patients still resist these immunotherapies. Present on CD8^+^T cells, immune checkpoint receptors are expressed by innate lymphoid cells (ILCs), which may contribute to the clinical response. ILCs are composed of natural killer (NK) cells, which are cytotoxic effectors involved in tumor immunosurveillance. NK cell activation is regulated by a balance between activating receptors that detect stress molecules on tumor cells and HLA-I-specific inhibitory receptors. Helper ILCs (h-ILCs) are newly characterized ILCs that secrete cytokines and regulate the immune homeostasis of tissue. We investigated the modulation of blood ILCs in melanoma patients treated with ipilimumab. Circulating ILCs from metastatic stage IV melanoma patients and healthy donors were studied for their complete phenotypic status. Patients were studied before and at 3, 6, and 12 weeks of ipilimumab treatment. A comparison of blood ILC populations from donors and melanoma patients before treatment showed changes in proportions of ILC subsets, and a significant inverse correlation of CD56^dim^ NK cells and h-ILC subsets was identified in patients. During treatment with ipilimumab, percentages of all ILC subsets were reduced. Ipilimumab also impacted the expression of the CD96/TIGIT/DNAM-1 pathway in all ILCs and increased CD161 and CTLA-4 expression by h-ILCs. When considering the response to the treatment, patients without disease control were characterized by higher percentages of CD56^bright^ NK cells and ILC1. Patients with disease control displayed larger populations of activated CD56^dim^CD16^+^ DNAM-1^+^ NK cells, while anergic CD56^dim^CD16^−^DNAM-1^−^ NK cells were prominent in patients without disease control. These results provide original findings on the distribution of ILC subsets in advanced melanoma patients and their modulation through immunotherapy. The effects of ipilimumab on these ILC subsets may critically influence therapeutic outcomes. These data indicate the importance of considering these innate cell subsets in immunotherapeutic strategies for melanoma patients.

## 1. Introduction

Melanomas are highly metastatic tumors for which the first-line treatment is immunotherapy. Immune checkpoint blockers targeting CTLA-4 and PD-1 restore the T lymphocytes antitumor reactivity and induce long-lasting responses [1,2,3,4]. However, the vast majority of patients develop either primary or secondary resistances [5]. Natural killer (NK) cells are potent cytolytic effectors that play a role in the innate and adaptive antitumor immune responses. NK cells are endowed with the capacity to kill tumor cells without prior sensitization, and NK cells are considered highly sophisticated detectives of harmful changes in cells themselves and are pivotal catalyzers of adaptive T cell responses [6,7]. For instance, NK cells are supress considered the major source of IFNγ in vivo, and NK-derived IFNγ is crucial for priming T helper 1 responses [8]. NK cells originate from bone marrow and maturate in the lymph nodes. Though they mainly circulate, NK cells also infiltrate lymphoid organs and tissue by migrating through blood and lymphatics.

Blood NK cells represent 5 to 20% of circulating lymphocytes and contain two major sub-populations according to the expression level of the adhesion molecule CD56 (NCAM) and the expression of CD16 (FcγR): CD56^dim^CD16^+^ and CD56^bright^CD16^−^ subsets. The CD56^dim^ population predominates in blood (90% of NK cells) and at sites of inflammation, exhibits a high cytotoxic potential, and broadly expresses MHC-I-specific inhibitory receptors. In contrast, the CD56^bright^ subset predominates in lymph nodes (95% of NK cells), produces cytokines upon activation, displays a low cytotoxic potential, and is considered to be a precursor stage of terminal CD56^dim^ NK cells [8,9].

NK activation depends on an intricate balance between activation and inhibitory signals, which determine whether the target will be susceptible to NK-mediated lysis [10,11]. Three main natural cytotoxicity receptors (NCRs) involved in NK cell activation were identified: NKp46 and NKp30, expressed by resting NK cells, and NKp44, induced after stimulation by cytokines. Activation of NK cells is triggered by additional receptors. NKG2D, which is expressed by a majority of peripheral NK cells, binds MHC-related antigen (MIC)-A/B molecules and UL16-binding proteins (ULBP1-4), which are induced on the membranes of stressed cells. The DNAX accessory molecule 1 (DNAM-1), an adhesion molecule belonging to the immunoglobulin superfamily, promotes many of these functions in vitro. NK cells require DNAM-1 for the elimination of tumor cells that are comparatively resistant to the NK-cell-mediated cytotoxicity caused by the paucity of other NK-cell-activating ligands [12]. Simultaneous engagement of NKp46 and DNAM-1 induces cytotoxicity and cytokine secretion by resting NK cells [13]. NK cell activation is controlled by inhibitory HLA-I-specific NK receptors, KIR (killer Ig-type receptor), which are present in the NK CD56^dim^ subset, and the ubiquitous C lectin-type CD94/NKG2A receptor, which binds to HLA-E molecules [14]. The KIR receptors CD158a and b recognize HLA-Cw4 (C2 type) and HLA-Cw3 (C1 type) molecules, respectively [15,16].

Activation of endogenous NK cells with interleukin (IL)-2 and adoptive transfer of in vitro activated autologous NK cells mediate antitumor activity in experimental and clinical settings [17]. NK cells can cure human melanoma lung metastases in nude mice treated with chronic indomethacin therapy combined with multiple rounds of IL-2 [18]. Studies of immune infiltrates in melanoma from The Cancer Genome Atlas (TCGA) indicate that patients with brisk tumor infiltrating lymphocytes (TIL) scores had improved melanoma-specific survival. They further outlined an association between cytokine/chemokine expression and immune tumor infiltrates, suggesting that the expression of specific tumor cytokines (IFNγ, IL10, TGFβ) represents important biomarkers of melanoma immune response [19].

Recently, NK cells were classified in the family of innate lymphoid cells (ILCs), which gather cytotoxic effectors (NK cells) and helper-type ILCs (h-ILCs) [20]. The h-ILCs arise from a common lymphoid progenitor [21,22] and are located in mucosal barriers and various tissues [23], including the intestine, skin, lung, and adipose tissue [24]. ILCs are heterogeneous subsets of lymphocytes that have emerging roles in orchestrating immune response and contribute to maintaining homeostasis and regulating tissue inflammation [25]. Compared to NK cells, h-ILCs are rare in blood relative to other immune cell populations. They are classified into distinct subsets—ILC1, ILC2, and ILC3—which mirror the subsets of CD4^+^ helper T cells. ILC1 produces IFNg, ILC2 produces IL5 and IL13, and ILC3 produces IL17 and IL-22. ILC subsets have distinct cytokine and transcription factor profiles that align with their biological functions. However, it has emerged that ILC subsets are not phenotypically fixed, but exhibit a considerable heterogeneity and plasticity in different environments [26,27].

The role of h-ILCs in disease pathogenesis has been reported, and their involvement in tumor immunosurveillance is suspected in growing numbers of studies [28]. Several distinct subsets of h-ILCs at skin barriers are involved in the complex regulatory network in local immunity, potentiating adaptive immunity and the inflammatory response [29]. Recent research progress in h-ILC biology has outlined abnormal functions and potential pathogenic mechanisms in autoimmune-related skin diseases. The diversity and plasticity of ILCs and their unique functions in disease conditions suggest their potential value in immunotherapy [30].

A prerequisite for the intelligent implementation of NK cells in an antitumor regimen is a thorough understanding of the changes in the different ILC subsets in tumor patients compared to donors and monitoring in response to immunotherapy. Here, we investigated the phenotype and function of blood ILCs in a series of metastatic melanoma patients before and during the course of immunotherapy.

Discrete blood h-ILCs are found in humans, and taking into account their proximity to NK cells, it is interesting to consider these different subsets in cancer patients receiving immunotherapy. The present study assesses the circulating innate lymphoid populations in a new manner, revealing significant changes in the different ILC subsets in blood from melanoma patients and the impact of ipilimumab on these populations.

## 2. Patients and Methods

### 2.1. Patient Characteristics and Study Design

Patients with unresectable stage III or IV melanoma were eligible for ipilimumab treatment if they had previously failed in one or more systemic therapies. Ipilimumab is a fully humanized anti-CTLA-4 IgG1k monoclonal antibody approved by the food and drug administration (FDA) for late-stage melanoma after having been demonstrated to extend overall survival in two pivotal phase III trials in a subset of patients [1,31]. Patients received intravenous ipilimumab in doses of 3 mg/kg every 3 weeks for an expected total course of four doses. Treatment was discontinued if either a related adverse event that could not be tolerated occured or rapidly progressive disease was noted. Blood samples from 16 patients were collected before each injection of ipilimumab at weeks 0, 3, 6, 9, and 12. All patients gave their informed consent for this study in accordance with institutional ethical guidelines. Blood samples from 12 healthy donors were collected from the blood donor center (Etablissement Français du Sang, Hôpital Saint-Louis).

### 2.2. Definition of Tumor Assessment and Main Clinical Outcomes

Tumors were staged with total-body computed-tomography (CT) scans and cerebral magnetic resonance imaging (MRI) at the baseline, week 12, and week 16. The overall response was assessed according to the immune-response-related criteria (irRCs) derived from time-point response assessments (based on tumor burden).

### 2.3. Flow Cytometry and Monoclonal Antibodies

Absolute lymphocyte counts were determined using the TruCount system (BD Biosciences) with CD45 PerCPb mAbs (BD Multitest, BD Biosciences, Franklin Lakes, NJ, USA).

Immunophenotyping and ILC analyses were performed on frozen peripheral blood mononuclear cell (PBMC) samples. PBMCs were first stained with Flexible Viability Dye eFluor 506 (eBioscience, San Diego, CA, USA) for 30 min and then stained with surface antibodies (Appendix A) in brilliant Stained Buffer (BD) at 4 °C for 30 min. Cells were fixed and acquired on LSRFortessa (BD) and analyzed with FlowJo10.

Total blood ILCs were identified as lin^−^ CD7^+^ cells. CD56 and CD16 further distinguished NK cells, while CD127, CD117, and CRTH2 were used to recognize helper ILCs. Markers in lin included CD3, CD4, CD8, CD5, TCR, CD33, CD19, CD14, and CD235a to exclude T cells, B cells, erythrocytes, and myeloid cells. NK cells were gated as CD56^+^CD127^-/low^CD16^+/−^, ILC1 as CD127^+^CD56^−^CD117^−^CRTH2^−^, ILC2 as CD127^+^CD56^−^CD117^+/−^CRTH2^+^, and ILC3 as CD127^+^CD56^−^CD117^+^CRTH2^−^. In addition to this backbone, the expression of NK receptors (NCR, NKG2D, CD161, NKG2A, NKG2C, DNAM-1, CD96), markers of activation (CD69, CD62L), and immune checkpoints (CTLA-4, PD-1, TIGIT) was determined in ILC subpopulations. A comparison between donors’ and patient’s blood ILCs was analyzed. In addition, we compared blood ILC subsets in patients before and after 3, 6, and 12 weeks of ipilimumab treatment.

### 2.4. CITRUS Analysis

The CITRUS (cluster identification, characterization, and regression) algorithm (Cytobank) was used to compare the abundance of ILC (lin-CD7^+^) subsets and expression of markers between donors and patients, as well as between disease controller and non-controller patients. CITRUS uses a hierarchical clustering map based on similarities in the expression of markers (Appendix A). The minimum cluster size of events of the total data used was 2%, and the number of events sampled per file was 5000. The analysis was based upon the abundance of events using significance analysis of microarrays (SAM), with a false discovery rate of 0.01.

### 2.5. Statistical Analyses

The data are described using means and standard deviations for quantitative variables and counts and percentages for qualitative variables. Comparisons between patients’ and donors’ (HD) immunological statuses at baseline were performed using a Mann–Whitney test (Prism 8, GraphPad Software Inc., San Diego, CA, USA). Comparisons between different times of treatment for the same patient were performed using a Wilcoxon rank sum test. Correlations were performed with R studio using the non-parametric Spearman correlation test. A *p* value < 0.05 was considered significant.

## 3. Results

### 3.1. Melanoma Patients Exhibit a Specific Distribution of Blood ILCs

The proportions of ILC subsets were comparable in donors and patients before treatment (Figure 1A). The NK cells constituted the major subsets, including >80% of CD56dim and 1–12% of CD56bright NK cells. Among the h-ILCs, which represented 2 to 12% of the CD7^+^ cells, there was a major subset of ILC1 (2 to 6%) and 2–4% of ILC2; ILC3 cells were rare (mean value of <0.5% of lin^−^CD7^+^ cells). There was a trend of increasing percentages of ILC2 in patients compared to donors (Figure 1B). We performed correlation analyses to determine the relations between the proportions of the different ILC subsets. In melanoma patients, the percentages of CD56^dim^ cells were inversely correlated with the percentages of ILC1, ILC2, and ILC3 (coefficient correlation > 0.7). In patients, the three h-ILC subsets were positively correlated with each other, while ILC2 and ILC3 did not correlate in donors (Figure 1C). These data suggest that in patients with metastatic melanoma, the relationships between the NK subsets and h-ILC subsets were altered, suggesting a change in the homeostatic regulation of these populations in the blood.

### 3.2. Phenotypic Analysis of ILC Subsets in Donors and Patients

A comparison of the ILC populations in donors and patients before treatment was performed with the CITRUS (cluster identification, characterization, and regression) algorithm, which was designed for the fully automated discovery of statistically significant stratifying biological signatures. Using multiparametric flow cytometry data, it allows the identification of ILC subsets that are significantly different in donors and patients before treatment (Figure 2A). CITRUS identified two NK populations that were less present in patients’ blood. These populations included activated CD56^+^CD16^+^NKp46^+^NKG2D^+^NKG2A^−^ and CD56^+^CD16^++^NKp46^+^NKG2D^low^ NK cells. Supervised analyses of discrete h-ILC populations showed that blood ILC1 cells had a phenotype that resembled the one of NK cells expressing NK-activating receptors, NKp30 and NKG2D, and inhibitory TIGIT and CTLA-4. These NK-activating receptors were not expressed by ILC2 and ILC3. ILC2 expressed DNAM-1 and CTLA-4; ILC3 was NKR negative and exhibited high CTLA-4 expression (Figure 2B). A comparison of donors’ and patients’ ILCs showed similar phenotypes of ILC subsets before ipilimumab treatment. There were modest differences, which included an increase in CD69 and decreased CD62L expression by all ILC subsets in melanoma patients (Figure 2B).

### 3.3. Ipilimumab Induces Early Changes in ILC Proportions

Ipilimumab induced early changes in the proportions and the phenotypes of the ILC subsets. First, the percentages of lin^−^ CD7^+^ cells were significantly reduced at 6 weeks (W6) after the beginning of treatment (Figure 3A). The percentages of the ILC1, ILC2, CD56^bright^, and CD56^dim^ subsets were decreased through the treatment with ipilimumab (Figure 3B).

Phenotypic analyses of ILC subsets were performed at W12 of the treatment (Figure 3C), revealing modest phenotypic changes. The CD96/DNAM-1/TIGIT pathways were altered in patients. In treated patients, CD96 expression by all ILCs was significantly reduced, except for ILC3 (*p* = 0.08). There was a trend of decreased expression of DNAM-1, whereas TIGIT was not changed. In addition, there was a higher expression of CD161 by ILC1 and ILC2 in the treated patients. The low CTLA-4 expression in ILCs was increased in ILC1 cells and reduced in ILC3. Expression of NKp46 was reduced and increased in the ILC3 subsets from ipilimumab-treated patients (Figure 3C).

### 3.4. Correlation with Clinical Status and Response to Treatment

The clinical status of the 16 patients included was evaluated at W16 of treatment; eight patients had progressed, corresponding to patients without disease control (w/oDC), and eight had stable or reduced tumors, corresponding to patients with disease control (DC).

First, we characterized absolute lymphocyte counts (ALCs), as lymphopenia is a frequent feature of cancer patients with advanced disease associated with poor prognosis. In a previous study [32] including a larger series of patients, peripheral T cell lymphopenia was associated with poor disease control, and ALC > 10^9^/L before treatment was correlated with survival and clinical response to ipilimumab. Moreover, T cell counts increased early (W3) following ipilimumab treatment and were maintained at W12. In the present study, none of the 8 DC patients were lymphopenic, and all had increased ALC in response to ipilimumab (Figure 4A). In contrast, three out of seven of the patients w/oDC were lymphopenic, and ipilimumab had no effect on the ALC. We found no correlation between absolute ILC numbers and disease control and no change in ILC counts following treatment with ipilimumab (Figure 4A, right).

The proportions of blood ILC subsets differed between the two groups of patients (Figure 4B, lower panels). Before treatment, we found a trend of higher percentages of CD56^bright^ NK cells and a significant increase in ILC1 percentages in w/oDC patients than in patients with DC, whereas the CD56^dim^ NK cell percentages were comparable. Ipilimumab affected the proportions of ILC subsets in patients w/oDC, reducing the percentages of CD56^bright^, CD56^dim^ NK cells, and ILC1 at W12 compared to at D0. Ipilimumab did not induce any significant changes in h-ILC proportions in patients with DC (Figure 4B).

A comparison of the ILC populations in patients with DC and in w/oDC patients was performed with CITRUS, thus allowing the identification of ILC subsets that distinguished the two groups of patients (Figure 5). Before (Figure 5A) and at W12 of treatment (Figure 5B), patients with DC were characterized by elevated abundance of CD56^dim^/CD16^+^/DNAM-1^+^/TIGIT^+^ NK cells, while NK CD56^dim^/CD16^−^/DNAM-1^−^ NK cells were prominent in patients w/oDC. Thus, patients with DC were characterized by lower percentages of ILC1 and the presence of mature activated CD56^dim^ NK cells.

## 4. Discussion

Immunotherapies that use immune checkpoint blockers (ICBs) are efficient treatments for metastatic melanoma patients. However, their efficiency is limited to some patients, and yet numerous patients either resist or relapse. If the reinvigoration of T cell cytotoxicity is the most relevant mechanism underlying antitumor response in responding patients, other immune effectors, which are rarely investigated in immunomonitoring studies, may be involved. The family of ILCs includes helper-type ILCs, which are mainly resident in tissue and control immune homeostasis. Through the secretion of cytokine, they may be involved in the immunosurveillance of tumors. The ILC1 helper type shares the ability to produce IFNγ and TNFα with NK cells and is controlled by tbet. ILC2 is present in the skin and produces type 2 cytokines, including IL4 and IL13. ILCs rapidly respond to changes in their environment and are able to acquire new secretion capacities. As ILCs, including h-ILCs, are additional players in tissue immunosurveillance, here, we investigated these ILC populations in a series of metastatic melanoma patients that received ipilimumab.

The definition of ILCs implicates that the gating of NK cells is different from the one that is used for specific monitoring of NK cells. Indeed, ILCs are defined as lin^−^CD7^+^ cells, and CD56/CD16 or CD127/CD117/CRTH2 identify NK cells or h-ILCs, respectively. Among blood ILCs, NK cells largely predominated and h-ILCs represented discrete populations, with a main subset of ILC1 and very low proportions of ILC3. While the proportions of ILCs were similar in donors and patients, distinct correlations between blood-derived NK cells and h-ILCs in HD and in patients were observed. In donors, the NK and h-ILC subsets were independent, while in melanoma patients, there was a strong negative correlation between CD56^dim^ and h-ILCs. The skin contains all ILC subsets, with a predominance of ILC2. In cutaneous melanoma, there is an enrichment of NK, ILC1, and ILC3NCR^+^ cells. Moreover, NK cells and ILC3 were increased in liver and lung metastases, and ILC2 was increased in lung metastases [33]. In melanoma patients, the dynamic differentiation of ILCs in the tumor environment may allow circulation of other ILC subsets [26].

The most significant phenotypic difference between donors and patients was the reduction of activated NK cells in patients. The comparison of donor and patient blood ILCs using CITRUS outlined a decreased abundance of activated CD56^+^CD16^+^NCR^+^ with moderate or high NKG2D expression and no or low NKG2A expression. The downregulation of activation receptors on the CD56^+^CD16^+^ NK subsets from metastatic melanoma patients confirmed previously described numerous studies [34], including ours [35,36] on monitoring NK cells. Immunosuppressive factors and cytokines that are produced by the tumor as soluble molecules or presented on extracellular vesicles (EVs) may contribute to these immune defects of ILCs [33]. Furthermore, NK-derived extracellular vesicles can compromise NK cell function [37]. The higher expression of CD69 and lower CD62L in patients indicate that these ILCs may correspond to recirculating tumor-exposed ILCs [38]. The low CD62L can also indicate their altered cytotoxic function [39,40] and/or capacity to spill out of the tumor and migrate to lymph node.

Our results are in accordance with those of previous studies, showing that ipilimumab affected the ALC, with a sustained increase in T cell counts from W3 to W12 [32]. In contrast, ipilimumab did not affect the absolute ILC counts, but altered the blood proportions and the phenotypes of ILCs. Ipilimumab reduced the proportions of h-ILC subsets at W3, which remained low at W12. The proportions of NK CD56^dim^ were decreased at W3 and W6, but the values at W12 were comparable to values before treatment (D0). As ILCs rapidly adapt to changes in their environment, the impact of ICBs in the tumor lesions may account for these dynamic changes in the blood ILCs.

The proportions CD56^bright^ NK cells were reduced in treated patients at W12 compared to D0. Previous studies reported a change in NK cell subtype in melanoma patients treated with ICBs. Low baseline levels of NK cells and CD56^dim^ NK cells were correlated with a positive response to ipilimumab, and patients with low amounts of CD56^dim^ NK cells showed a trend of longer survival [41]. While percentages of CD56^dim^ were not reduced in patients with DC, non-supervised CITRUS analyses showed that higher percentages of CD56^dim^CD16^−^ were associated with patients w/oDC. Furthermore, the relative abundance of CD56^dim^CD16^+^ NK cells over CD56^dim^CD16^−^ NK cells characterized patients with DC before and after treatment with ipilimumab.

Thus, the most prominent ILC subset that was significantly overrepresented in patients with DC was that of CD56^dim^CD16^+^ NK cells. As CTLA-4 expression by NK cells and h-ILCs was weak, it excluded a direct effect of ipilimumab on NK cells. However, IgG1 antiCTLA-4 mAb can engage FcRIII on NK cells and favor ADCC-mediated lysis. It was reported that ipilimumab engaged CD16 ex vivo on non-classical monocytes by inducing ADCC-mediated lysis of regulatory T cells (Tregs) [42]. Melanoma cells express CTLA-4 [43], and the engagement of CTLA-4 on primary melanoma cell lines induced ADCC and TNFα production by NK cells [44]. Moreover, it was reported that an immunoconjugate between ipilimumab and EGFR (epidermal growth factor) targeting breast cancer cells activated NK cells, and inhibited tumor cells more efficiently than each compound alone [45]. The presence of CD56^+^CD16^+^ NK cells and ADCC may have a prognostic value for ipilimumab treatment outcomes.

In melanoma patients, h-ILC and CD56^dim^ were inversely correlated, and h-ILC1 cells were associated with clinical response to ICBs. Low h-ILC1 percentages were present in patients with DC, while larger proportions of ILC1 characterized melanoma patients that did not benefit from ipilimumab, suggesting that ILC1 exerts pro-tumor immune functions. Accordingly, the conversion from NK to ILC1-like cells that produced VEGF (vascular endothelial growth factor) was recently reported, leading to metastasis development [46]. In addition, TGFβ was shown to induce the conversion of NK cells into ILC1, which lacked antitumor capacities [47]. At the baseline, ILC2 frequencies tended to be higher than in donors. Increased numbers of ILC2 were already found in gastric cancers and chronic lymphocytic leukemia [48,49]. In several reports on lung metastasis models, IL-33-activated ILC2s were shown to mediate a potent immunosuppression of NK cells, promoting metastasis [50,51]. On the other hand, ILC2s were reported to amplify the antitumor effect of PD-1 blockades by activating tumor-specific T cells in an orthotopic murine model of pancreatic tumors [52], emphasizing the requirement for characterization of intra-tumoral ILC populations [33].

In addition to CTLA-4 and PD-1, additional checkpoint molecules involved in tumor immunosurveillance target ILCs. A comparison of DC and NR patients with CITRUS outlined a high expression of DNAM-1 in DC patients. The involvement of NCR and DNAM-1 in the antitumor function of NK cells against melanoma cells was reported earlier [53]. DNAM-1/CD155 interactions promoted cytokine production and NK-mediated suppression of metastasis of poorly immunogenic murine melanoma [54]. DNAM-1 (CD226) binds several ligands, including CD155 (poliovirus receptor, PVR), which also associate with the inhibitory receptor TIGIT and with CD96; the latter receptor leads to a controversial effect on NK cell function and activity [55]. Soluble ligands (sCD155) also altered DNAM-1 signaling [56,57]. IL15 stimulation with TIGIT blockades reverses altered CD155-mediated NK cell function in melanoma [58]. High CD155 expression by tumor cells was associated with PD-1^+^ CD8^+^T cells and poor response to anti-PD-1, and CD155^+^PD-L1^−^ tumors exhibited a poor response to anti-PD-1/CTLA-4 therapy [59]. Preclinical studies suggest that targeting this axis can improve immune-mediated tumor control, particularly when combined with existing anti-PD-1 checkpoint therapies [60]. Combined targeting of CD96 and PD-1 receptors potently reduced lung metastases in murine models. An anti-CD96 antibody that inhibited the CD96/CD155 interaction stimulated the proliferation and local production of IFNγ by NK cells. The targeting host CD96 was shown to complement conventional immune checkpoint blockades [61] and further outlined the relevance of this pathway for NK-cell-based immunotherapy. Another newly developed ICB, anti-NKG2A (Monalizumab), which blocks CD94/NKG2A expressed by mature NK cells and T cell subsets, exerted antitumor activity as a single agent or in combination with anti-PDL1 or anti-EGFR [62].

Overall, the immune checkpoint inhibitors that alter the tumor microenvironment change the crosstalk between resident ILCs, leading to anti- or pro-tumoral responses [33]. Our studies identify innate immunity parameters in the blood melanoma patients that are correlated with a response to immunotherapy.

## Figures and Tables

**Figure 1 cancers-13-01446-f001:**
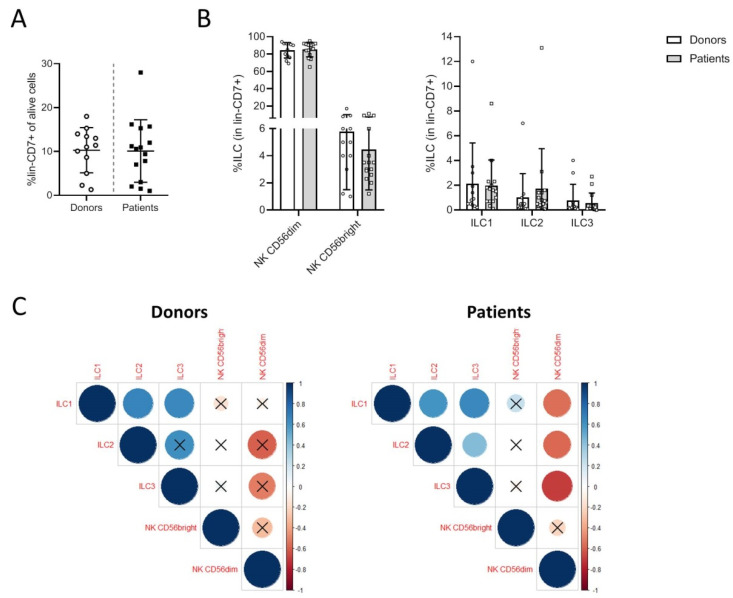
Proportions of innate lymphoid cell (ILC) subsets among the global lin-CD7+ population. Percentages of lin^−^CD7^+^ ILCs in live peripheral blood mononuclear cell (PBMC) from donors and patients (**A**). Percentages of the different ILC populations in lin^−^CD7^+^ cells (**B**). Correlations between the proportions of ILC populations in lin^−^CD7^+^ cells. A Spearman test was used to calculate the correlation coefficient (X: *p* value > 0.05) (**C**).

**Figure 2 cancers-13-01446-f002:**
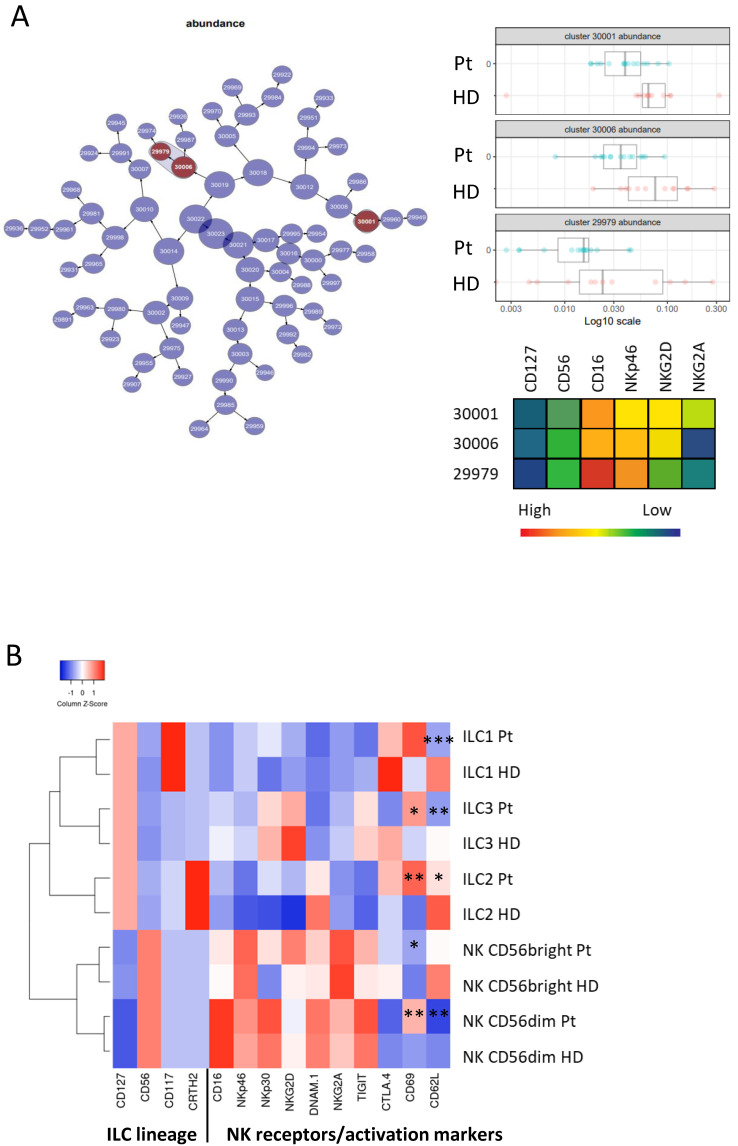
Phenotypic characteristics of blood ILC subsets in donors’ (HD) and patients’ (Pt) blood. Hierarchical clustering map of the CITRUS (cluster identification, characterization, and regression) analysis for comparing donors and patients before treatment. Clusters whose cell abundances are different between the two conditions are in dark red. Cell abundance in the three clusters is represented on the left side with a heatmap presenting the phenotypes of these clusters (**A**). Heatmap presenting the markers expressed by ILC populations in HD and in Pt. Mann–Whitney test was used to compare HD and Pt (* *p* < 0.05, ** *p* < 0.01, *** *p* <0.005) (**B**).

**Figure 3 cancers-13-01446-f003:**
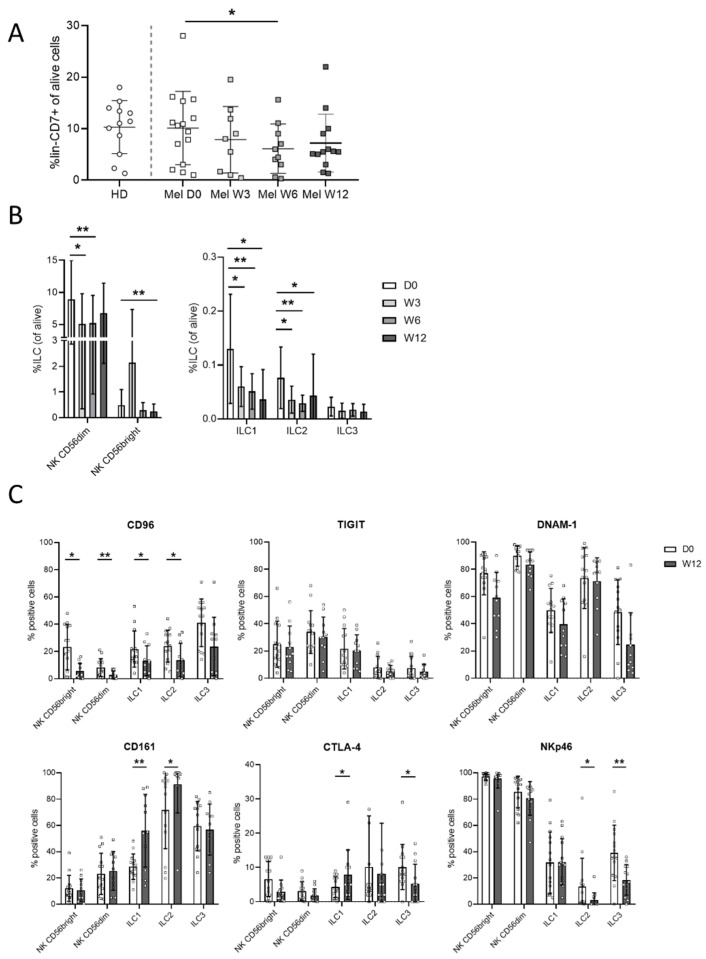
Phenotypic assessment of the immune checkpoint in ILCs from patients treated with ipilimumab. Percentages of lin^−^CD7^+^ ILCs in live PBMCs from donors and patients following ipilimumab treatment at day 0, weeks 3, 6 and 12 (D0, W3, W6, W12) (**A**). Percentages of the different ILC populations in lin^−^CD7^+^ cells in patients following treatment (**B**). Expression of receptors in ILC populations in patients at D0 and at W12 of treatment (**C**). A Wilcoxon matched-pair signed rank test was used to compare percentages between D0 and other times of treatment (* *p* < 0.05, ** *p* < 0.01).

**Figure 4 cancers-13-01446-f004:**
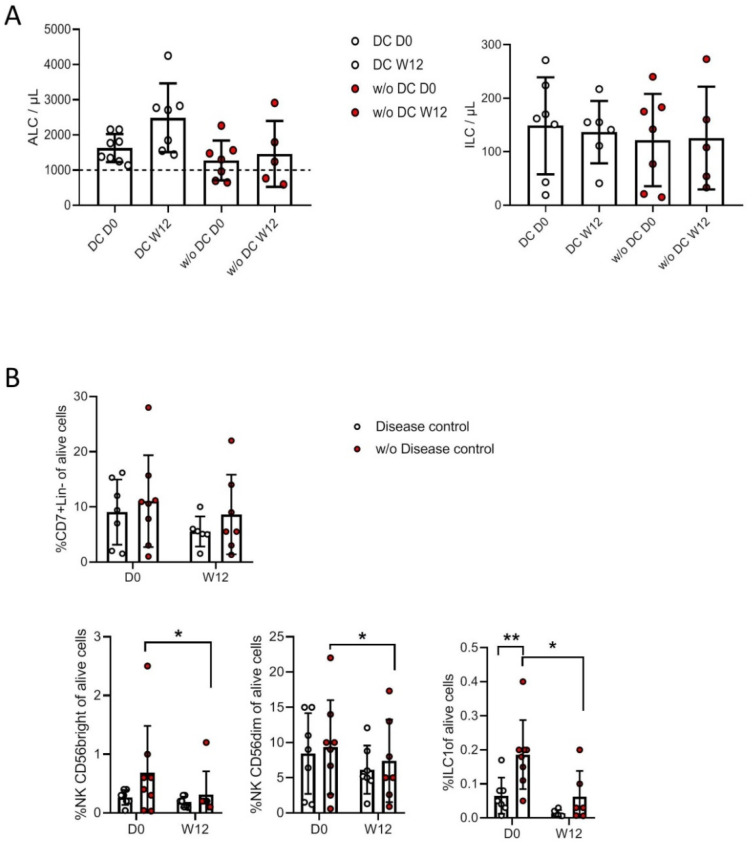
Assessment of ILC subsets in patients with disease control (DC) and in patients without disease control (w/oDC). Absolute lymphocyte counts (ALC) and absolute ILC counts per µL in DC and w/oDC patients (**A**). Percentages of lin^−^CD7^+^ ILC, CD56^bright^, CD56^dim^, and ILC1 in live PBMCs from DC and w/oDC patients (**B**). A Wilcoxon matched-pair signed rank test was used to compare the percentages in patients with DC between D0 and other times of treatment. A Mann–Whitney test was used to compare the percentages between DC and w/oDC patients (* *p* < 0.05, ** *p* < 0.01).

**Figure 5 cancers-13-01446-f005:**
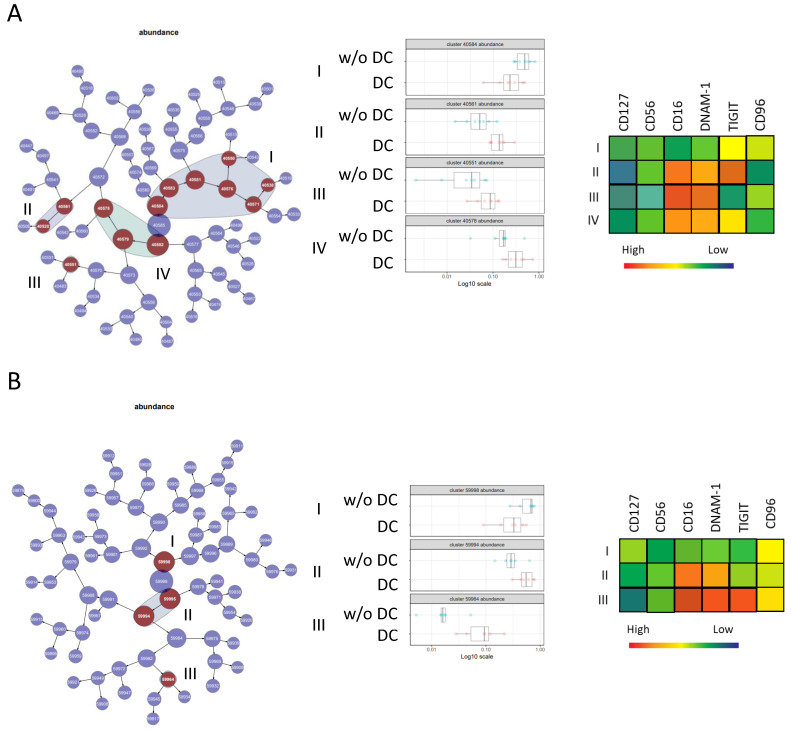
Comparison of natural killer (NK) subpopulations in DC patients and in patients w/oDC with citrus. A hierarchical clustering map of the CITRUS analysis comparing DC patients and w/oDC patients at D0 (**A**) and W12 (**B**). Clusters whose cell abundances are different between the two conditions are in dark red. Four groups of clusters are noted at D0 and three at W12. The cell abundance in the different clusters is represented on the left side with a heatmap presenting the phenotypes of these clusters.

## Data Availability

All data associated with this study are present in the paper or in the Appendix A.

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
