# Peer review of "Specific Patterns of Blood ILCs in Metastatic Melanoma Patients and Their Modulations in Response to Immunotherapy"

_cancers, 2021, doi:10.3390/cancers13061446_

Round 1
Reviewer 1 Report
The manuscript presented by Rethacker et al, entitled: “Specific Patterns of Blood ILC in Metastatic Melanoma Patients and Their Modulations in Response to Immunotherapy” provides new insights on the distribution of innate lymphoïd cells subsets in advanced melanoma patients and their modulation by immunotherapy.
The Authors have investigated the modulation of blood ILC in melanoma patients treated with Ipilimumab, and shown that the effects of the treatment on these ILC subsets may influence therapeutic outcomes. The analysis of innate cells subsets in melanoma patients represents an important step for effective immunotherapeutic strategies.
The data shown here is novel, well presented, and it will make an important advancement in the field of melanoma management.
Author Response
We thank the reviewer for the positive comments on the manuscript
Reviewer 2 Report
Main findings of the study:
This study was undertaken to clarify the modulation of ILCs in melanoma patients treated with the immune checkpoint inhibitor ipilimumab. This is a well written manuscript, with results that may have influence on our therapeutic considerations. The methods used are valid and mostly suitable to answer the proposed questions. Results and data are interpreted with appropriate caution.
While this is an interesting manuscript, I have a major and several minor concerns.
My major concern is the following: The most important drawback of this study is that it investigates ILCs in peripheral blood samples. As stated also by the authors themselves, ILCs mainly reside and exert their functions in tissues. Therefore, it remains a question how relevant these results could be regarding the in situ anti-melanoma functions of ILCs. It would have been interesting and perhaps more informative to see similar results from tumor biopsies.
Minor concerns:
Lines 194-197 - The authors state that „Supervised analyses of discrete h-ILC populations showed that blood ILC1 cells had a phenotype close to NK cells expressing NKR, NKp30 and NKG2D and inhibitory TIGIT and CTLA-4. Activating NKR were not expressed by ILC2 and ILC3. ILC2 express DNAM-1 and CTLA-4 and ILC3 were NKR negative and exhibited high CTLA-4 expression.” According to the heatmap in Figure 2B, the phenotypes of ILC1 and NK cells are different, and the NK receptors are expressed on ILC3 cells. How do you explain this?
Lines 227, 237, 241 – The authors use confusing terminology for patients without disease control, and present different numbers: first 8 patients w/o DC are mentioned, while later 7 NR (supposedly non responder) patients are listed. Please clarify!
Please discuss the available evidence on the effect of immune checkpoint inhibitors on ILCs.
The authors should cite and discuss the following article: Aintzane Apraiz et al: Innate Lymphoid Cells in the Malignant Melanoma Microenvironment. Cancers 2020, 12, 3177; doi:10.3390/cancers12113177.
There are a few typos that needs to be corrected, e.g. line 276: ‘g’ and ‘a’ instead of γ and α; line 279: ‘inclusing’.
Overall, I recommend the publication of the article with after these modifications.
Author Response
Thanks to the reviwer's comments, we submit an improved revised version of our manuscript.
A point by point relpy to each comment of the reviewer is joined

Reviewer 3 Report
Thank you for your submisison.
This is an unteresting article with novel findings and also the graphs and strings show importante evidences. I advice only to add in the discussion some periods about Natural killer derived extracellular vesicles, which are constitutively secreted and biologically active, contributing to immunosurveillance and first-line defense in the control of tumor growth and metastasis diffusion. In this regard add this reference: Natural-Killer-Derived Extracellular Vesicles: Immune Sensors and Interactors. Front Immunol. 2020 Mar 13;11:262. doi: 10.3389/fimmu.2020.00262. PMID: 32231660; PMCID: PMC7082405.
Author Response
We thank the reviewer for the postive comments on our manuscript. In the revised version, we have discussed the importance of NK derived extracellular vesicles in the immunosurveillance of tumors and added the reference from Federici C et al, Front Immunol, 2020.
Reviewer 4 Report
The authors describe differences in ILC subtypes in healthy donors and patients with metastatic melanoma that received ipilimumab. Furthermore, they compared the ILC subsets of responders and non-responders. The methods are clearly described; however, it is not clear what the clinical benefit of the results is, e.g. whether the detection level of certain ILCs can predict response.
Detailed comments:
- Large parts of the introduction are lacking references (e.g. section from line 37 to line 49; sections from line 84 to line 103 and until the end of the introduction).
- Methods: Is there a number from the ethics review board available?)
- First section of results (line 159-169 rather belongs to the methods)
- Line 172ff: % of these subsets are hard to understand because the graph showes the proportions in a different way. Text should reflect the graph.
- Line 177: …were positively correlated “with each other”…
- Figure 1B: please provide p-values. Is anything significant here?
- Line 199: what does “close phenotype” mean? Similar phenotype?
- Figure 2B: p-values are missing in the legend
- Figure 3A: How do you explain that values are not significant anymore at W12?
- Figure 3B: why huge increase at W3? There is no significant decrease among W0 W6 W12. Also dim population: increase again at W12 -> not significant anymore. Why?
- Figure 3C: line 215 It says CD96 was reduced in all ILC, however it is not significant in ILC3. Same for line 218: it is only significantly reduced in ILC3, anything else is not significant.
- Figure 3C: What is the value in healthy donors for all these markers? Do they differ from D0 melanoma patients?
- Line 228: two spaces between patients and had
- Figure 4A left: p-values? You mention an increase of ALC in DC patients. Is anything significant?
- Line 239: should point out “Figure 4A, right”.
- Figure 4B and line 247ff: The graphs are not mentioned in the correct order in the text.
- Line 255: words are in a different style and size
- Line 254-260: It is not clear from the text what figure 5 shows. A and B is not mentioned in the text.
- Line 269: ICB abbreviation was not introduced before
- Line 279: “including”?
- Line 284ff: This is a repetition of the results. What is the explanation/conclution?
- Line 309f: “Low baseline 309 levels of NK cells and CD56dim NK cells correlated with a positive response to ipilimumab” This is not shown in the results. Baseline of CD56dim is similar in DC and w/o DC (Figure 4B).
Author Response
Thanks to the reviewer's comment, we submit an improved revised version of our manuscript
A point by point reply to each comment of the reviewer is joined

Round 2
Reviewer 4 Report
Thank you for improving the manuscript.